# Peelable Nanocomposite Coatings: “Eco-Friendly” Tools for the Safe Removal of Radiopharmaceutical Spills or Accidental Contamination of Surfaces in General-Purpose Radioisotope Laboratories

**DOI:** 10.3390/pharmaceutics14112360

**Published:** 2022-11-01

**Authors:** Traian Rotariu, Daniela Pulpea, Gabriela Toader, Edina Rusen, Aurel Diacon, Valentina Neculae, John Liggat

**Affiliations:** 1Military Technical Academy “Ferdinand I”, 39–49 George Cosbuc Boulevard, 050141 Bucharest, Romania; 2Faculty of Chemical Engineering and Biotechnologies, University ‘POLITEHNICA’ of Bucharest, 1–7 Gh. Polizu Street, 011061 Bucharest, Romania; 3Institute for Nuclear Research—RATEN ICN-Pitesti, 1 Street Campului, 115400 Mioveni, Romania; 4Department of Pure and Applied Chemistry, University of Strathclyde, 295 Cathedral Street, Glasgow G1 1BX, UK

**Keywords:** radiopharmaceutical spill, accidental contamination, radionuclides, radioisotope laboratory, decontamination, nanocomposite coating, peelable/strippable films

## Abstract

Radioactive materials are potentially harmful due to the radiation emitted by radionuclides and the risk of radioactive contamination. Despite strict compliance with safety protocols, contamination with radioactive materials is still possible. This paper describes innovative and inexpensive formulations that can be employed as ‘eco-friendly’ tools for the safe decontamination of radiopharmaceuticals spills or other accidental radioactive contamination of the surfaces arising from general-purpose radioisotope handling facilities (radiopharmaceutical laboratories, hospitals, research laboratories, etc.). These new peelable nanocomposite coatings are obtained from water-based, non-toxic, polymeric blends containing readily biodegradable components, which do not damage the substrate on which they are applied while also displaying efficient binding and removal of the contaminants from the targeted surfaces. The properties of the film-forming decontamination solutions were assessed using rheological measurements and evaporation rate tests, while the resulting strippable coatings were subjected to Fourier-transform infrared spectroscopy (FTIR), thermogravimetric analysis (TGA), differential scanning calorimetry (DSC), and tensile tests. Radionuclide decontamination tests were performed on various types of surfaces encountered in radioisotope workspaces (concrete, painted metal, ceramic tiles, linoleum, epoxy resin cover). Thus, it was shown that they possess remarkable properties (thermal and mechanical resistance which permits facile removal through peeling) and that their capacity to entrap and remove beta and alpha particle emitters depends on the constituents of the decontaminating formulation, but more importantly, on the type of surface tested. Except for the cement surface (which was particularly porous), at which the decontamination level ranged between approximately 44% and 89%, for all the other investigated surfaces, a decontamination efficiency ranging from 80.6% to 96.5% was achieved.

## 1. Introduction

The primary goal of radiopharmacy in nuclear medicine is to use radiolabeled compounds for diagnostic [1] and therapeutic [2] purposes. Because the treatment frequently targets life-threatening diseases, the benefits of using radionuclides exceed the risks. Thus, for many decades, radiopharmaceutical-based therapy has been used to treat cancer [3]. Radiotherapy benefits from radioisotopes with short physical half-lives, ranging from a few hours to a few days. Radionuclides used in radiopharmaceutical therapy are mostly beta or alpha particle emitters [4]. Several radionuclides, such as ^131^I, ^32^P, ^90^Sr, ^90^Y, have been employed to treat many benign and malignant disorders [5].

In the radiopharmacy domain, working with radiation from open sources entails persistent responsibility. Given this, contamination from radioactive spills may occur more frequently than in other areas of the nuclear medicine department or other radioactive facilities. The eventual contamination of the working surfaces augments personnel radiation exposure and might increase the probability of the contamination of their hands, face, or equipment. Moreover, the contamination of surfaces is a major concern because it may facilitate cross-contamination of the products handled in the radiopharmacy. Hence, contamination represents a severe threat to the health of exposed personnel, and its mitigation is the best straightforward method of minimizing a hazard [6,7,8].

In general, an accidental release of radionuclides, which implies the discharge of less than 100 microcuries (3.7 × 10^6^ Bq) of a radioisotope found in a nonvolatile form, can be considered a ‘minor’ incident. In comparison, the release of radionuclides in a nonvolatile form exceeding 100 microcuries (3.7 × 10^6^ Bq), or the discharge of any amount of a radionuclide in a volatile state, or spilling of a high volume of radioactive materials, must be considered a ‘major’ contamination event [9].

Soap and water mixtures, conventional chelators (e.g., EDTA), alkaline solutions, bleach, or alcohol are common, inexpensive cleaners that ensure a reasonable degree of decontamination [6,10,11,12]. Additionally, several formulations designed for the decontamination of radionuclides are commercially available, in a variety of forms: powdered formulations (e.g., Alconox^®^), sprayable liquid solutions (e.g., Radiacwash™, COUNT-OFF™), scrubbing mixtures (e.g., GritMitts^®^), foams (e.g., TFD Mousse^®^ Franklab), aerosols (e.g., NoCount^®^), etc. These products may provide the desired results, but their major shortcoming is represented by the volume of waste they generate thru the decontamination process, which requires additional, costly, disposal procedures thereafter.

Along with the aforementioned products, a distinct class of decontamination products is represented by strippable coatings [13]. They usually consist of viscous mixtures that can be sprayed or applied by brush on the contaminated surface. Their major advantages are their ability to generate peelable films that can be subsequently easily disposed of and a considerably lower amount of post-decontamination waste. The commercial strippable formulations that are already on the market (e.g., DeconGel^®^ 1108, ALARA™ 1146) may promise high decontamination effectiveness and generate fewer residues. Still, they contain solvents or other toxic volatile components [14], which require additional precautions for the personnel performing decontamination. In addition, not all the polymers used in these formulations are readily biodegradable. At the same time, many studies in the literature proposed various solutions for developing strippable coatings for radionuclides decontamination purposes. A series of relevant studies for this domain are cited below.

H.N. Gray et al. described the synthesis of “smart” polymeric coatings [14] for surface decontamination based on poly(vinyl alcohol) and poly(vinyl pyrrolidone), poly(ethylene oxide), poly(vinylamine) or poly(ethenyl formamide) (PEF), as polymeric matrices, and ethylenediaminetetraacetic acid disodium salt (EDTA) or diethylenetriaminepentaacetic acid (DTPA), as chelating agents, and demonstrated the successful removal of uranium and plutonium. Yang et al. developed sprayable polyvinyl alcohol–borax complex formulations [15], which can be removed after decontamination by rinsing with water, containing various types of adsorbents (Prussian blue, bentonite, and sulfur-encapsulated chabazite), and obtained a removal efficiency of only 56.96% for ^137^Cs. H.M. Yang et al. described the synthesis of polyacrylamide/alginate/Fe_3_O_4_ peelable hydrogel films [16] for ^137^Cs, obtaining a maximal removal efficiency of approximately 96% from paint-coated cement. Xu et al. described a film-forming detergent obtained from ethyl cellulose (EC), tea polyphenols (TP), polyvinyl acetate (PVAc), and polytetramethylene ether glycol bis-para-aminobenzoate (P1000). They reported a decontamination rate of approximately 85% for uranyl acetate [17].

Unfortunately, most of these methods have significant drawbacks, including the use of dangerous chelating agents (such as EDTA [14]), the large amounts of radioactive secondary liquid waste generated during the rinsing step [15] (that necessitate highly-priced water post-treatment), or the fact that they turn into supplementary radioactive waste after use if the polymeric matrices employed for entrapping the contaminants are not readily biodegradable [13,16,18]. Additional examples of some of the major limitations of the currently available formulations designed to remove radionuclides include poor radionuclide entrapment efficacy [14], excessive generation of high amounts of post-decontamination waste [15], adverse effects of their components on the health of the operators performing decontamination [6,7], or negative impact on the environment [19]. Even though few of the currently available techniques have demonstrated their ability to remove radionuclides, further research is still necessary to improve the following aspects: decontamination performance, improving the suitability of the strippable coatings for distinct decontamination scenarios, enabling the multifunctionality of these materials, ensuring the safety of the personnel that performs decontamination, and also improving the environmental friendliness of strippable coatings [17,19,20].

In this context, to address the shortcomings of the existing solutions, the present study aimed to develop novel ecological nanocomposite coatings for the entrapment and removal of radionuclides from contaminated surfaces. At the end of the decontamination process, these peelable coatings, containing the entrapped contaminants, can be easily compacted and stored in small containers dedicated to radioactive waste. Since these matrices are entirely biodegradable, their degradation products will not require subsequent disposal. Thus, to the best of our knowledge, this is the first time a research study envisaged decontaminating formulations comprising exclusively ‘green’ components, that can be used as an alternative to the existing solutions. Owing to their ecological composition and design, these formulations can significantly reduce the volume of waste generated after decontamination, while still ensuring a high decontamination efficiency. These innovative, inexpensive, ‘eco-friendly’, strippable nanocomposite films, designed for radionuclides decontamination, were obtained from water-based, non-toxic, formulations comprising readily biodegradable [21,22,23] components: polymeric matrices (gelatin, sodium alginate, polyvinyl alcohol), a new generation [12,24,25] ‘green’ chelating agent (iminodisuccinic acid), glycerol, and bentonite nanoclay. These new formulations are suitable for the decontamination of the surfaces (e.g., workbenches, sinks, gloveboxes, flooring, etc.) existent in general-purpose radioisotope laboratories (radiopharmacy, nuclear medicine, nuclear research, etc.) or for the mitigation of radioctive leakages/spills in nuclear facilities (e.g., nuclear power plants, industrial irradiation facilities). For this purpose, a series of six novel decontamination solutions were synthesized, and their efficiency in entrapping and removing ^90^Sr-Y (as representative for β−emitting radionuclides) and ^241^Am (as representative for α−emitting radionuclides) from various types of surfaces (concrete, painted metal, ceramic, linoleum, epoxy resin) was demonstrated. 

## 2. Materials and Methods

### 2.1. Materials

Poly(vinyl alcohol) (PVA, average Mw ≈ 85,000–124,000 Da, 87–89% hydrolyzed, Sigma Aldrich), sodium alginate (Alg, white powder seaweed extract from Special Ingredients Ltd., Glasgow, UK), gelatine (Glt, GELITA^®^ Pharmaceutical Gelatin, Limed Bovine Bone Gelatine, 250 bloom, 8 mesh), iminodisuccinic acid (IDS, BAYPURE^®^ CX 100, Lanxess), hydrophilic bentonite (BT, Nanomer^®^ PGV, Sigma–Aldrich), glycerol (Gly, purity ≥99.5%, Sigma–Aldrich), were used as received. Radionuclides solutions for controlled contamination consisted of radioactive solutions of ^90^Sr-^90^Y (β−emitting representative radionuclide) and ^241^Am (α−emitting representative radionuclide), in 0.1N HNO_3_, initial activity: ^90^Sr/Y—1.64 kBq/g and ^241^Am—600 Bq/g. Surfaces employed for decontamination tests: concrete (C, approximately dimensions 12 mm × 12 mm × 0.8 mm), painted metal (PM, approximately dimensions 20 mm × 10 mm × 0.2 mm), ceramic tiles (CT, approximately dimensions 8 mm × 8 mm × 0.5 mm), linoleum (L, linoleum for flooring purposes, approximately dimensions 10 mm × 10 mm × 0.3 mm) and epoxy resin (P, ProLab^®^ epoxy cover for workbenches, approximately dimensions 20 mm × 10 mm × 0.6 mm). It is important to note that none of the surfaces utilized in the decontamination test had been previously used, yet they had been wiped with alcohol before contamination.

### 2.2. Methods

#### 2.2.1. Synthesis of the Decontamination Solutions

The composition of each decontamination solution prepared is illustrated in Table 1. The synthesis of DS1 started with the dissolution of IDS in distilled water, followed by the dispersion of BT in this solution, with an ultrasonic processor (Sonics^®^ Vibracell VCX750W), for 30 min. Subsequently, PVA was added to the mixture and stirred with a high-shear disperser (IKA T18 digital ULTRA-TURRAX^®^) until it was completely dissolved. For DS2–DS6 preparation, three stock solutions of poly(vinyl alcohol) (PVA, 10 wt.% solution in water), sodium alginate (Alg, 2 wt.% solution in water) and gelatine (Glt, 5 wt.% solution in water) were prepared, and the appropriate amounts were combined with an aqueous solution (containing only IDS and BT), to obtain the concentrations indicated in Table 1. The final step for all decontamination solutions consisted of the addition of glycerol, followed by another 30 min of stirring the entire formulation. Thus, six film-forming decontamination solutions were prepared. 

#### 2.2.2. Controlled Contamination, Decontamination Procedure, and Activity Measurements

Each surface employed for the decontamination survey (illustrated in Appendix A: concrete, painted metal, ceramic, linoleum, epoxy resin) was contaminated with a total amount of 100 mg (divided in 6 droplets) of radioactive solution (described in Materials section). After being deposited on the targeted surfaces, the radioactive solutions were allowed to evaporate overnight. The next day, the activity of each specimen was recorded before applying the decontamination solution. Five specimens from each type of contaminated surface (Appendix A) were grouped, according to the contaminant utilized, in an aluminum Petri dish, and the decontamination solution was applied afterwards (Appendix A). The coatings were allowed to dry at room temperature. The next day, the nanocomposite coatings were peeled from the tested surfaces (Appendix A), and the activity of each specimen was recorded again (Appendix A). It is important to note that all measurements were conducted in triplicate and that the mean values were presented.

#### 2.2.3. Characterization

For the six decontamination solutions, dynamic viscosity was measured at 25 °C, with an IKA ROTAVISC me-vi instrument, equipped with a coaxial cylinder system (O-DINS-1), and controlled via ‘Labworldsoft^®^ 6 Visc’ dedicated software. The shear rate between 1 and 1300 s^−1^ was applied for all measurements. The evaporation rate of water contained in each decontamination solution was assessed by using an ATS 120 Axis Thermobalance. For this purpose, 4 g of each decontamination solution was poured in Petri dish with a diameter of 55 mm, maintained at constant temperature, and weighted at every 150 s, for 2 h. Triplicate measurements were performed for each DS, for each of the following temperatures: 25 °C, 30 °C, 35 °C, or 40 °C, and the mean values were reported. The peelable films were evaluated with an ATR modulus (PIKE MIRacle™) on a Bruker VERTEX 70 Spectrometer, to obtain the FTIR spectra. Data was collected by averaging 32 scans from 500 to 4000 cm^−1^ at a resolution of 4 cm^−1^. The thermogravimetric analyses (TGA) were executed with a Netzsch TG 209 F3 Tarsus instrument (Erich NETZSCH GmbH & Co. Holding KG, Selb, Germany), setting the following parameters: nitrogen flow rate was maintained at 20 mL/min, mass of the samples placed in alumina crucibles was approximately 3 mg, the temperature range for each analysis was 25–900 °C, with a heating rate of 10 °C/min. DSC analysis was performed on a Setaram Setline DSC equiment, on approximatively 10 mg of sample, from −25 °C to 300 °C, with a heating rate of 10 °C/min. The mechanical resistance of the thin nanocomposite films was investigated on a Discovery 850 DMA TA Instruments, by utilizing tensile test clamp for uniaxial deformation. Tensile tests were carried on rectangular specimens (50 mm × 8 mm × approximately 0.1 mm), at 5 mm/min. Tensile tests were performed on 5 specimens from each material, and the mean values were reported. The evaluation of the decontamination performances was estimated based on the data collected with a SPAB-15™ probe dedicated for the measurement of surface contamination, equipped with a Silicon PIPS^®^ detector with 15 cm^2^ detection area, an ideal tool for the direct measurement of alpha and beta emitters, provided with a Colibri TTC survey meter for dose-rate information. The SPAB-15™ probe is a completely integrated subsystem that collects and transmits measurements to the instrument utilized for display. The measurement range provided by this system ranged from 0 to 10,000 c/s, energy range—β >100 keV, α >3 MeV, background—β <0.8 c/s, α < 0.01 c/s. Each experiment was performed in triplicate, and the mean values were utilized to estimate the values of the decontamination efficiency for each formulation and type of surface tested. Thus, based on the activities recorded before and after each decontamination experiment, decontamination efficiency (*DE* [%]) was calculated according to the following equation: $$DE=\frac{A_{0}-A_{f}}{A_{0}}\;\cdot \;100,$$
where *A*_0_ represents the initial activity (measured before applying the decontamination solution) and *A_f_* represents the final activity (measured after peeling the nanocomposite coating).

## 3. Results and Discussions

### 3.1. Evaluation of the Properties of the Decontamination Solutions

The first part of this research consisted in the synthesis and characterization of the decontaminating formulations (DS). The ratio between the polymeric components of these aqueous blends was adjusted in accordance with Table 1 (Materials section), to evaluate the influence exhibited by the components from each DS on the properties of the resulting analogous nanocomposite coatings and, subsequently, on their decontamination performances. In one of our previous studies [24], we demonstrated that a solution containing 10 wt.% PVA and distinct chelating agents was efficient for the decontamination of several radionuclides, but the major inconvenience of PVA is represented by its low biodegradation rates [26,27]. Therefore, for achieving a higher level of eco-friendliness, the main goal of this study was to reduce the amount of PVA in the decontamination formulations, by replacing it with polymers of natural origin with higher biodegradability. Gelatine and alginate were selected for this purpose because they are readily biodegradable [28,29,30]. Thus, at the end of the decontamination process, the polymeric films containing the entrapped radionuclides can be more properly disposed, without generating additional waste. The major challenge faced in the synthesis of these new formulations (comprising gelatin, alginate, and less PVA) was represented by maintaining the mechanical resistance of the resulting strippable films, while still ensuring high decontamination efficiencies. In this context, from all the formulations obtained in the preliminary study, only six representative film-forming decontamination solutions were further investigated and herein reported (Table 1). The minimal concentration of PVA, which still ensured the proper mechanical resistance of the strippable coatings, was 3 wt.%, because the films obtained when PVA was completely absent from the decontaminating formulation had a tendency to become brittle and to be difficult to peel (DS6). Additionally, inspired by one of our previous works [20], we introduced hydrophilic bentonite nanoclay in all DS, in an attempt to enhance the decontamination abilities of these formulations designed for radionuclide removal, while also achieving a higher mechanical resistance of the strippable film. It is well-known that one method for enhancing some of the features of biodegradable polymers, such as the thermal, mechanical, and oxidative barrier, is employing low percentages of inorganic fillers. Bentonite is one of the most frequently employed lamellar silicates, as an inorganic filler, in conventional composites, because it is affordable, readily available, and environmentally friendly [31]. Glycerol was also added to the nanocomposites as a plasticizer to provide additional elasticity and to facilitate the peeling process.

Following the selection of six relevant DS (Table 1) for this survey, the following stage of this study consisted in the evaluation of two of their key parameters, which directly influence the decontamination process: viscosity and film-forming timing.

The viscosity of DS dictates the proper application method on the surface to be decontaminated: lower viscosity solutions are easily applied by spraying technique, while solutions with higher viscosity can be applied by brush or roller. Furthermore, higher viscosity may reduce the capacity of DS to enter the pores of the contaminated surface and, at the same time, it reduces the diffusion process, resulting in poorer decontamination performances [20]. As a result, the dynamic viscosity of the six DS was measured while increasing the shear rate. The results are comparatively displayed in Figure 1. All the investigated formulations exhibited a shear-thinning character. This behavior is specific for non-Newtonian fluids in which the fluid viscosity decreases with increasing shear stress [32]. To better distinguish the differences between the samples, the shear rate was plotted in a logarithmic scale. Therefore, it can be observed that DS6 and DS5 possess higher viscosities, due to higher polymeric content and most likely also due to the additional hydrogen bonds established between their components. DS2 possesses a higher viscosity than DS3, hence the influence of alginate on the increase of DS viscosity is slightly higher than in the case of gelatine (DS3). This influence of sodium alginate in augmenting the viscosity of DS is also observable when comparing samples DS4 and DS5. The highest viscosity values were measured when PVA was absent from the decontaminating formulation (DS6), and 1 wt.% of each of the two natural polymers (Alg and Glt) was employed.

Another important feature of the decontaminating formulations is represented by the coating formation rate directly related to the solvent evaporation rate (water, in our case). This process is directly influenced by temperature, relative humidity, evaporation area, and sample volume. For this experiment, we tried to keep each of these variables constant, except for the temperature, which was increased with each new experiment, to see how this parameter speeds up the coating formation. The results obtained for these experiments, at four distinct temperatures, are illustrated in Table 2 (triplicate measurements were recorded and the mean values were reported).

### 3.2. Evaluation of the Properties of the Strippable Nanocomposite Films

After assessing the main properties of the decontamination solutions, the subsequent step consisted in the characterization of the resulting nanocomposite films. All the nanocomposite peelable films reported in this study were obtained via a conventional casting method. The first investigation performed on the peelable nanocomposite films was FTIR characterization to examine the interactions established between the components of the decontaminating formulation. FTIR spectra are illustrated in Figure 2. The first broad signal at 3300 cm^−1^ in DS1 can be attributed to PVA hydrogen bonded O–H stretching vibrations. This signal appears slightly shifted towards 3290 cm^−1^ for the following samples (DS2–DS5), most likely due to the polymeric association via additional hydrogen bonds established between the main formulation and the new components introduced in DS (Alg and Gel). For DS6 the peak maxima situated around 3291 cm^−1^ could be attributed to N-H and O-H stretching in gelatine. C-H stretching is visible at 2945 cm^−1^ from DS1, appearing slightly shifted to 2940 cm^−1^ in the other polymeric blends (DS2–DS6). In FTIR spectra of DS3 and DS6, the peak specific for amide group (N-H bending, associated C=O stretching) can be observed at 1650 cm^−1^. In DS4, this peak is slightly shifted to 1640 cm^−1^ due to the accompanying interactions established between Alg and Gel. Sodium alginate exhibits significant carbonyl stretching vibration [33] at 1610 cm^−1^ as it can be noticed in the DS2 and DS5 spectra. C-O stretching is marked by the presence of the sharp peak at 1087 cm^−1^. The sharp peaks at 1026 cm^−1^ and the shoulder peak at 1030 cm^−1^ can be assigned to the Si–O stretching vibration from the bentonite, which are slightly shifted (from 1015 cm^−1^ in pure bentonite [20]) due to the Si-OH interactions with the polymeric blend (PVA, Alg, Gel). O–Si–O stretching [34] can be attributed to the peaks situated at 920 cm^−1^.

Further, TGA and DSC investigations were carried out to evaluate the thermal characteristics of the strippable coating, and the results can be seen below.

Thermogravimetric analysis (TGA) of the nanocomposite peelable coatings was conducted to ascertain the thermal resistance of the materials. Decomposition and oxidation reactions, or physical processes such as sublimation, evaporation, and desorption, can lead to weight changes in polymeric materials [35]. Figure 3a,b illustrates the TGA and DTG curves obtained for the nanocomposite films, and Table 3 contains information regarding the decomposition onset and maximum decomposition temperatures. The evaporation of water, which was still present after curing and drying in the polymeric nanocomposite films due to the hydrophilic nature of all the components, is associated with the initial weight loss observed before 100 °C [36]. In addition, a three-stage thermal degradation pattern was observed for most of the analyzed samples (DS1–DS4), except DS5 and DS6, for which a four-stage thermal degradation profile and a two-stage thermal degradation profile, respectively, were noticed. These transitions are more evident in the analogous peaks in DTG plots (Figure 3b). The first region between 50 and 150 °C can be attributed to the loss of absorbed water, while the next stage is attributed to water bound to the polymer matrix (up to 340 °C) while the third region is associated with the decomposition and carbonization of the polymer matrix [37].

According to TGA results, it seems that the thermal stability of these formulations was slightly increased by the presence of sodium alginate (DS2) or gelatine (DS3), in comparison with DS1, which can be explained by the slightly higher thermal resistance of sodium alginate [38] and gelatine [39]. In the case of alginate containing samples, the fourth weight loss region can be attributed to the weight loss registered by decomposition of the Na_2_CO_3_ obtained during the decomposition of the sodium alginate [40].

The DSC method was used for evaluating the thermophysical properties of the samples, as shown in Appendix A and Table 4. The second heating-cooling cycle was used to avoid thermal history effects. The step transition indicating the T_g_ of the crystalline regions of PVA was identified around 80 °C [41], and it is slightly shifted with +/− 5 °C, according to the composition of the films. DS6 does not contain PVA, therefore this step transition is not present in this plot. The second transition observed was ‘β-relaxation’ which is described in the literature as T_β_, the relaxation in the PVA crystalline domains [41]. Thus, the broad peaks between 90 and 170 °C can be ascribed to the crystallinity changes in the structure of the polymeric blend [20,41,42]. A sharp peak was noticeable only in the samples without alginate, DS1 and DS3, between 170 and 180 °C, suggesting that these two polymeric blends possess a greater amount of highly crystalline regions, or more ordered regions inside the polymeric matrix. The melting of the PVA crystalline regions [41] may also be responsible for the transitions visible on the DSC curves in 170–225 °C interval, but it is also clear that after 210 °C the components of the polymeric blend progressively decompose.

The next investigation performed was the uniaxial deformation tensile test. The strippable coatings used for decontamination purposes must be flexible and elastic, therefore the mechanical properties of the films are crucial for enabling coating removal from the contaminated surfaces. Thus, specimens from each type of nanocomposite film were subjected to the uniaxial tensile test for evaluating their mechanical resistance. Five specimens from each material were tested, but, for the ease of visualization only, the average results are illustrated in Figure 4. As can be observed in Figure 4 and Table 5, DS2, DS4, and DS5 exhibited both elasticity and resistance to higher loadings, in comparison with DS1 and DS3, which demonstrated similar elasticity but lower stress resistance. In contrast, DS6 was stiff and fragile, and this can also be observed from the tensile test results. Thus, to create strippable coatings that can be readily peeled off from decontaminated surfaces without cracking, it is crucial to know how each component affects the mechanical properties of the coatings when formulating film-forming decontamination solutions. Therefore, in our situation, tensile test results allowed us to confirm that all formulations from DS1 to DS5 are suitable to be employed as peelable coatings, except DS6, which is a brittle film and is not suitable for this particular purpose.

### 3.3. Evaluation of the Decontamination Efficiency

Figure 5 illustrates a decontamination scenario involving an accidental spilling of a solution containing radionuclides. The principle of strippable coatings decontamination implies the use of a film-forming solution, which can be applied on the contaminated surface, and possesses the ability to entrap and thus to remove the radionuclides from the contaminated surface. Once the decontamination solution is placed on the contaminated surface, it begins to interact with the contaminants through adsorption and complexation mechanisms, undertaken by the nanoclay adsorbent, sodium alginate, and the ‘green’ chelating agent (IDS), respectively. Thus, the contaminants are entrapped and ‘sealed’ inside the matrix of the nanocomposite. After the complete curing of the nanocomposite film, it can be simply removed by peeling and it can be compacted and safely stored in a small recipient for radioactive waste.

The main advantages of these decontamination formulations consist in their high decontamination efficiency, inexpensiveness, non-toxic nature, and ecofriendliness, generating almost no waste due to the readily biodegradable nature of the components. As mentioned above, their decontamination performances are guaranteed via multiple paths: the high chelating ability of IDS [24], the ability of alginate to bind multivalent cations [43,44], and the high adsorption capacity of bentonite [45,46].

Thus, after assessing the properties of the decontaminating formulations, the last objective of this research work was to evaluate and demonstrate their efficiency in entrapping and removing α and β emitter radionuclides from various types of surfaces found in general-purpose radioisotope laboratories. For this purpose, four of the six above presented decontamination solutions were employed (DS1, DS2, DS3, and DS5). To assess how the components, affect the decontamination performances, the selection was based on both their mechanical characteristics and composition.

For the controlled contamination stage, we employed two representative radioactive solutions, one for β—emission contamination (^90^Sr-^90^Y) and one for α—emission contamination (^241^Am). Even if for diagnosis applications ^241^Am is used in encapsulated pellets [47], in general-purpose radioisotope laboratories, contamination with actinides is still a matter of concern [48].

Thus, in an attempt of demonstrating the decontamination efficiency of the herein reported nanocomposite coatings, we performed decontamination tests, on five types of surfaces and by employing two types of radionuclide contaminants (as detailed in Methods section). The results obtained are illustrated in Figure 6 and Appendix A.

One of the major objectives of this study was to establish the adequate composition of the decontaminating formulation, for achieving maximal results in terms of decontamination efficiency. As can be observed from Figure 6 and Appendix A, except for the cement surface (which was highly porous), a decontamination efficiency ranging from 80.6% to 96.5% was achieved for all the other investigated surfaces. It is important to note that the radionuclides were successfully captured and removed by the decontamination solutions, even though we performed a controlled contamination with an acidic solution of radionuclides (0.1 N HNO_3_). As it can be observed from the comparative plots illustrated in Figure 6, the decontamination efficiency varies with the type of composition employed, the type of surface tested, and the type of contaminant. The highest decontamination efficiencies were obtained for ^90^Sr-^90^Y, especially on PM and CT while for ^241^Am the best results were obtained on linoleum surface.

In terms of decontamination efficiency on the same type of surface, the difference between the four decontamination solutions employed for this survey was not as significant as we expected, but for ^90^Sr-^90^Y it seems that DS5 offered the best results.

The decontamination process with the peelable coatings method is possible via two main mechanisms: complexation and adsorption [25]. Appendix A (from the Supporting info file) contains some hypothetical illustrations of the interactions that may be established between the contaminants and the complexing agents from the decontaminating solutions: the “green” chelating agent IDS [49] and sodium alginate. IDS acts as a pentadentate ligand (Appendix A), forming chelates having a octahedral structure [50] with the radionuclides, while the interactions established between the radionuclides and sodium alginate can be described by the thru “egg-box” model (Appendix A). Bentonite, besides improving the mechanical resistance of the strippable coatings, may enhance the decontamination efficiency through its outstanding ability to adsorb metallic ions [20,46,51]. Thus, when the aqueous decontamination solution is applied to the contaminated surface, the radionuclides are entrapped in the polymer-clay system via the above-described mechanisms. The other components of the decontamination solutions (PVA, Glt, Gly) facilitate the formation of a resistant, elastic, easy-peelable coating, suitable for safely removing the entrapped contaminants.

After analyzing the above illustrated results, we can affirm that the decontamination survey offered substantial information and a broad overview of the efficiency of these nanocomposite coatings in removing α and β emitter radionuclides from various types of surfaces.

## 4. Conclusions

This paper aimed at the synthesis and application of six types of film-forming decontaminating solutions for the removal of radionuclides, for dealing with accidental spillage on surfaces common in a nuclear facility. The obtained nanocomposite coatings were resistant and could be easily peeled off, efficiently removing the entrapped contaminant this way. Thus, after decontamination, the films can be safely disposed as radioactive waste.

These novel nanocomposite peelable coatings were obtained from water-based, non-toxic, polymeric blends containing readily biodegradable components, which do not harm the substrate on which they are applied, while also binding and removing radionuclides from the affected surfaces.

The new materials obtained were characterized through specific analytic techniques: for the decontamination solutions, the viscosity and evaporation rate were measured, while the resultant strippable coatings were subjected to FTIR, TGA, DSC, and tensile tests. Subsequently, various types of surfaces found in radioisotope workspaces were subjected to radionuclide decontamination studies (concrete, painted metal, ceramic tiles, linoleum, epoxy resin cover). Through the extensive investigations performed, it was shown that they possess remarkable properties (thermal and mechanical resistance, ease of strippability) and that their efficiency in entrapping and removing beta and alpha particle emitters depends on the constituents of the decontaminating formulation, but predominantly on the type of surface tested.

We presume that the main element driving these formulations to successful decontamination is the ‘green’ chelating agent employed (IDS), and the primary contribution of sodium alginate or gelatin to this achievement consists mainly in their positive effect on the mechanical performances of the nanocomposite. Nevertheless, each element of these decontaminating formulations plays a specific role (IDS—chelating agent, PVA, Alg, Gel—polymeric matrices, BT—adsorbent and reinforcing agent, Gly—plasticizer, water—solvent), contributing to the success of the decontamination process, which implies not only removing the radionuclides from the contaminated surface but also generating a nanocomposite film that ‘seals’ the contaminants, and which can subsequently be easily peeled of and safely disposed. For most of the tests, decontamination efficiencies obtained exceeded 90%, and for those which did not reach this decontamination level, a second application could be perhaps useful, and this could be the perspective of further investigations.

## Figures and Tables

**Figure 1 pharmaceutics-14-02360-f001:**
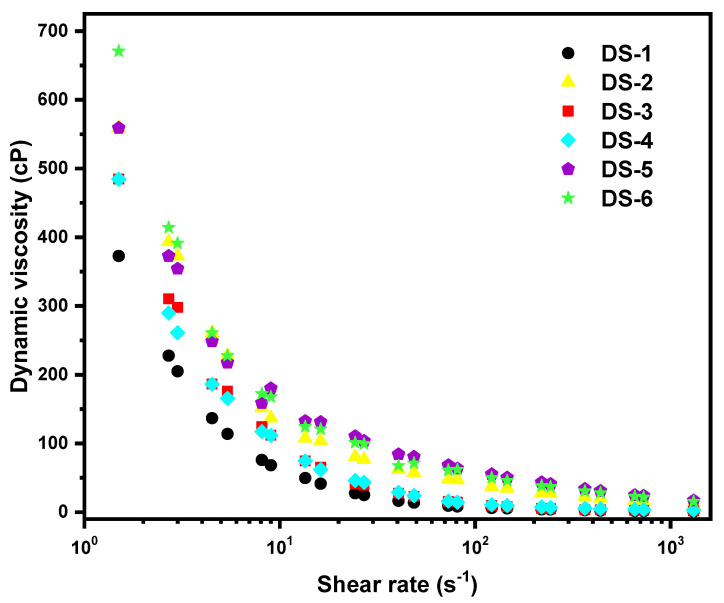
Viscosity of the decontaminating solutions.

**Figure 2 pharmaceutics-14-02360-f002:**
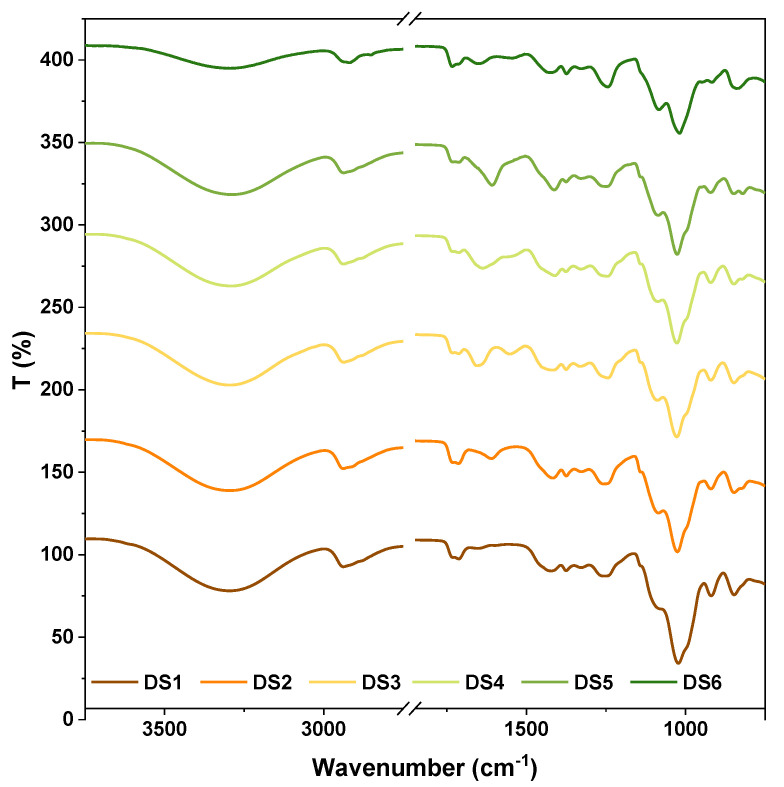
FTIR spectra of the nanocomposite films.

**Figure 3 pharmaceutics-14-02360-f003:**
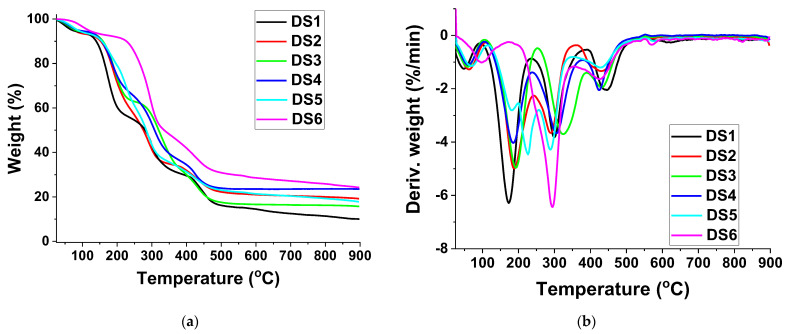
(**a**) TGA and (**b**) DTG curves obtained for the nanocomposite films.

**Figure 4 pharmaceutics-14-02360-f004:**
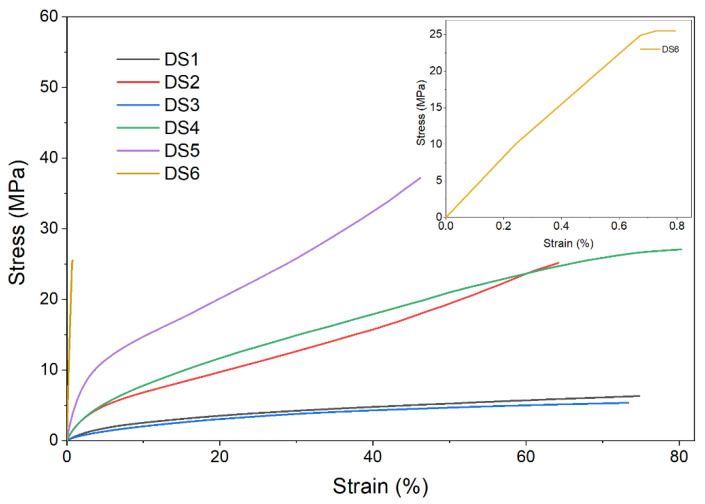
Stress–strain plots for the nanocomposite films.

**Figure 5 pharmaceutics-14-02360-f005:**
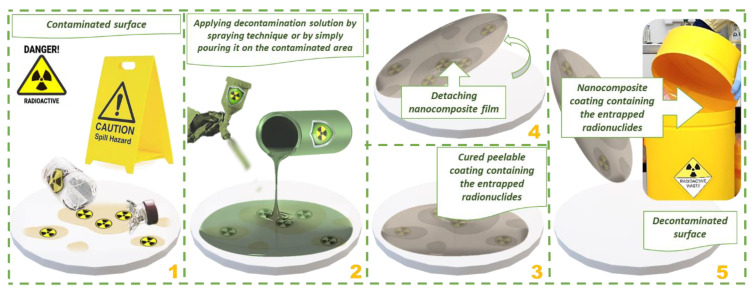
Radionuclide accidental spilling: decontamination scenario employing the nanocomposite strippable coatings. (**1**) Securing the contaminated area; (**2**) Applying the decontaminating formulation; (**3**) Entrapping the contaminants inside the polymer-clay system and allowing the curing of the nanocomposite film; (**4**) Peeling the film containing the entrapped radionuclides; (**5**) Safe disposal with minimal amount of waste.

**Figure 6 pharmaceutics-14-02360-f006:**
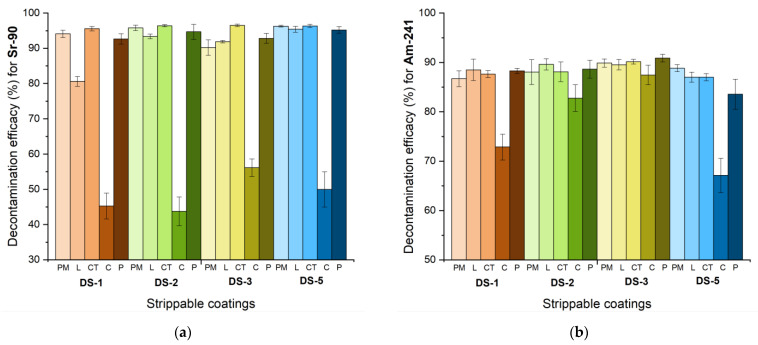
Decontamination efficiency obtained for (**a**) ^90^Sr-^90^Y and (**b**) ^241^Am.

**Table 1 pharmaceutics-14-02360-t001:** Composition of decontamination solutions.

Sample	Glt (wt.%)	Alg (wt.%)	PVA (wt.%)	IDS (wt.%)	BT (wt.%)	Gly (wt.%)
DS1	-	-	10	0.5	1	2.5
DS2	-	1	3	0.5	1	2.5
DS3	1	-	3	0.5	1	2.5
DS4	1	0.5	3	0.5	1	2.5
DS5	0.5	1	3	0.5	1	2.5
DS6	1	1	-	0.5	1	2.5

**Table 2 pharmaceutics-14-02360-t002:** Solvent evaporation rates for different compositions and temperatures.

Temperature	25 °C	30 °C	35 °C	40 °C	45 °C	30 °C	35 °C	40 °C
Sample	Evaporation Rate (mg/min)				Total Film-Forming Duration (h)			
DS1	4.4	7.1	10.3	15.8	12.7	7.8	5.3	3.5
DS2	3.3	6.7	11.0	15.4	16.6	8.2	5.0	3.5
DS3	4.2	7.6	10.8	14.8	13.3	7.2	5.0	3.7
DS4	3.5	6.7	10.8	15.0	15.6	8.1	5.0	3.6
DS5	4.3	7.3	11.3	15.9	12.9	7.4	4.8	3.5
DS6	3.8	7.2	11.8	16.0	14.9	7.8	4.8	3.5

**Table 3 pharmaceutics-14-02360-t003:** Thermal properties of the obtained nanocomposites DS1, DS2, DS3, DS4, DS5, and DS6.

Sample	T_10%_, [°C]	T_max_, [°C]
DS1	140	50, 175, 300, 445
DS2	152	64, 190, 290, 430
DS3	157	60, 192, 324, 430
DS4	155	60, 185, 305, 425
DS5	156	69, 181, 226, 289, 426
DS6	225	97, 295, 425, 570, 822

T_10%_ = decomposition onset temperature (measured at 10% weight loss); T_max_ = the maximum decomposition temperature, corresponding to the maximum of DTG peak (the first derivative of the thermogravimetric curve).

**Table 4 pharmaceutics-14-02360-t004:** Glass transition (Tg) temperatures for the obtained nanocomposites.

**Sample**	DS1	DS2	DS3	DS4	DS5	DS6
**T_g_ [°C]**	72.0	67.6	70.0	74.9	70.7	46.5

**Table 5 pharmaceutics-14-02360-t005:** Tensile test results for the nanocomposite films.

Sample	Young’s Modulus * (MPa)	σ_max_ (MPa)	ε_max_ (%)
DS1	0.58 ± 0.2	6.3 ± 0.4	74.8 ± 0.6
DS2	1.34 ± 0.6	25.1± 0.4	64.2 ± 0.8
DS3	0.41 ± 0.2	5.3 ± 0.5	73.3 ± 0.7
DS4	1.46 ± 0.2	27.0 ± 0.4	80.2 ± 0.6
DS5	4.74 ± 0.4	37.2 ± 0.5	46.2 ± 0.8
DS6	36.86 ± 1.2	25.5 ± 1.2	0.8 ± 0.4

* Young’s modulus was calculated in the range 0.1–3%, where the plot was considered linear.

## Data Availability

Not applicable.

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
