# Peer review of "Peelable Nanocomposite Coatings: “Eco-Friendly” Tools for the Safe Removal of Radiopharmaceutical Spills or Accidental Contamination of Surfaces in General-Purpose Radioisotope Laboratories"

_pharmaceutics, 2022, doi:10.3390/pharmaceutics14112360_

Round 1

Reviewer 1 Report

The manuscript "pharmaceutics-1996161" by Rotariu et al. reported the Peelable nanocomposite coatings: “eco-friendly” tools for the safe removal of radiopharmaceutical spills or accidental con-tamination of surfaces in general-purpose radioisotope labora-tories. After review, this study is interesting but the authors have to make major changes. The authors should refer to the following comments to improve their work:

General comments:

a. The language of the manuscript should be checked. There are many errors.

b. Please show in a schematic the interaction between materials. This can give a better understanding to the reader.

c. Use proper format for all tables.

d. Correct reference format according to journal.

Specific comments:

Abstract:

a. Terms such as FT-IR, TGA, and DSC are abbreviated in the abstract without explanation, please revise and rewrite.

Introduction:

a. The introduction follow a purposeful structure and content.

b. Please review the results of several other studies in the introduction and explain the difference between this study and the others.

c. Please move the explanation of EDTA from paragraph 6 to paragraph 4 (line 1).

Materials and methods:

a. The thickness of the layers (coatings) is not specified. Please, give more information about the case.

b. Why were these contents (0.5 and 1) chosen from Glt and Alg?

c. Please move Figures 2 and 4 to the Supporting Information.

Thermal section:

a. The results obtained in this section are very interesting, but unfortunately, there has been no discussion and it is only a data report, but these results can be used in the best way. I suggest that describe the mechanism of performance of the materials carefully.

b. In Figure 7b, the curves of samples cannot be distinguished from each other at all. Show the curves in distinct colors that are recognizable.

c. It is better to show the curves up to a temperature of 1000 °C so that the stabilization of the curves can be seen (especially the DS5 and DS6 curve).

d. List all temperature ranges in a table and move Figure 7 to the Supporting Information.

e. DSC analysis section:  
Please remove the sentence "Additionally, DSC analysis was performed to obtain complementary data regarding the thermal properties of the nanocomposite coatings. Thus, DSC thermograms are comparatively illustrated in Figure 8." and replace it with this sentence: "DSC method was used for evaluating the thermophysical properties of samples as shown in Figure 8."

f. In figure 8, vertical axis, please indicate the direction of exothermicity.

g. In Figure 8, DS6 curve, what is the reason for the peak appearing at 50°C?

h. In figure 8, DS5 curve, what is the reason for the peak appearing in the range of 0 to 25°C?

i. In figure 8, DS2 curve, what is the reason for the peak appearing in the range of -10 to -25°C?

j. Why the peaks that were asked in the previous three comments have appeared in the same curves and are not seen in the rest of the curves?

k. List all temperature ranges in a table and move Figure 8 to the Supporting Information.

Mechanical properties:

a. One of the most important sections of this manuscript is the mechanical properties. Important factors can be reported from the stress-strain curve, such as elongation at break, strength, and elastic modulus. Please report all these factors in a table and move Figure 9 to the Supporting Information. By presenting a table, comparison of samples will be done much better, and more accurate information will be given to the reader.

Decontamination section:

a. Please improve the discussion.

b. The quality of the Figure 11 is very poor and it is better to improve the quality.

Author Response

General comments:

a. The language of the manuscript should be checked. There are many errors.

The manuscript was checked, and errors were corrected.

b. Please show in a schematic the interaction between materials. This can give a better understanding to the reader.

A schematic illustration explaining the interaction between materials was added in Supporting info file.

c. Use proper format for all tables.

All tables have been formatted.

d. Correct reference format according to journal.

References are formatted now.

Specific comments:

Abstract:

a. Terms such as FT-IR, TGA, and DSC are abbreviated in the abstract without explanation, please revise and rewrite.

The manuscript was updated to include the abbreviations.

Introduction:

a. The introduction follow a purposeful structure and content.

The introduction was revised.

b. Please review the results of several other studies in the introduction and explain the difference between this study and the others.

The introduction part was improved with revised paragraphs and additional references were added.

c. Please move the explanation of EDTA from paragraph 6 to paragraph 4 (line 1).

The explanation was moved following the reviewer’s suggestion.

Materials and methods:

a. The thickness of the layers (coatings) is not specified. Please, give more information about the case.

Information is contained in the following sentence: „Tensile tests were carried on rectangular specimens (50mm × 8mm × approx. 0.1mm), at 5mm/min.

b. Why were these contents (0.5 and 1) chosen from Glt and Alg?

The concentrations chosen for Glt or Alg were incrementally increased : 0%, 0,5% and 1%, to evaluate their influence on the performances of the coatings. We have stopped at 1% because the coatings became brittle.

c. Please move Figures 2 and 4 to the Supporting Information.

Since Figures 1 – 4 can be better understood together, we have moved them all to Supporting info.

Thermal section:

a. The results obtained in this section are very interesting, but unfortunately, there has been no discussion and it is only a data report, but these results can be used in the best way. I suggest that describe the mechanism of performance of the materials carefully.

The TGA results discussion was improved following the reviewer’s suggestion.

  1. In Figure 7b, the curves of samples cannot be distinguished from each other at all. Show the curves in distinct colors that are recognizable.

Figure 7 (Figure 3 now) was modified (different colors were already used to distinguish the samples and we replaced dot-lines with solid lines), in accordance with the reviewer’s suggestion.

  1. It is better to show the curves up to a temperature of 1000 °C so that the stabilization of the curves can be seen (especially the DS5 and DS6 curve).

The DS5 and DS6 contain alginate which leads to the formation of Na2CO3 which undergoes thermal degradation at high temperatures (we have performed the TGA up to 900°C, since the maximum for our equipment is 1000°C). There is not a significant information to be gained if the range is increased by 100°C. All weight loss domains are already present in the analysis for the selected range (25-900°C).

d. List all temperature ranges in a table and move Figure 7 to the Supporting Information.

A table containing the information regarding the temperatures was added to the manuscript, but we have opted to keep Figure 7 (Figure 3 now) in the manuscript for a clearer presentation of the results.

  1. DSC analysis section:  
    Please remove the sentence "Additionally, DSC analysis was performed to obtain complementary data regarding the thermal properties of the nanocomposite coatings. Thus, DSC thermograms are comparatively illustrated in Figure 8." and replace it with this sentence: "DSC method was used for evaluating the thermophysical properties of samples as shown in Figure 8."

We have modified the sentences following the reviewer’s observation.

f. In figure 8, vertical axis, please indicate the direction of exothermicity.

We have updated Figure 8 (Figure 3 now)  and introduced the exo up sign.

g. In Figure 8, DS6 curve, what is the reason for the peak appearing at 50°C?

For DS6 the signal at around 50°C is an endothermic relaxation event, that can be attributed to the Tg of gelatin polymer.

  1. In figure 8, DS5 curve, what is the reason for the peak appearing in the range of 0 to 25°C?

The analyses were repeated, and the second cycle was used to remove the thermal stress in the material which was responsible for the initial signals present on the original graphs.

i. In figure 8, DS2 curve, what is the reason for the peak appearing in the range of -10 to -25°C?

The analyses were repeated and the second cycle was used to remove the thermal stress in the material which was responsible for the initial signals present on the original graphs.

  1. Why the peaks that were asked in the previous three comments have appeared in the same curves and are not seen in the rest of the curves?

This is probably due to alginate PVA interaction. The other samples did not display the same residual thermal stress.

k. List all temperature ranges in a table and move Figure 8 to the Supporting Information.

DSC plots were moved to Supporting info, and we added Table 4 (including the Tg values for the samples) to the manuscript.

Mechanical properties:

a. One of the most important sections of this manuscript is the mechanical properties. Important factors can be reported from the stress-strain curve, such as elongation at break, strength, and elastic modulus. Please report all these factors in a table and move Figure 9 to the Supporting Information. By presenting a table, comparison of samples will be done much better, and more accurate information will be given to the reader.

Table 5 was added, but we decided to keep the figure in the manuscript for a more facile comparison of the results.

Decontamination section:

a. Please improve the discussion.

The discussion was expanded.

b. The quality of the Figure 11 is very poor and it is better to improve the quality.

Figure 11 (Figure 6 now) was modified.

General comments:

a. The language of the manuscript should be checked. There are many errors.

The manuscript was checked, and errors were corrected.

b. Please show in a schematic the interaction between materials. This can give a better understanding to the reader.

A schematic illustration explaining the interaction between materials was added in Supporting info file.

c. Use proper format for all tables.

All tables have been formatted.

d. Correct reference format according to journal.

References are formatted now.

Specific comments:

Abstract:

a. Terms such as FT-IR, TGA, and DSC are abbreviated in the abstract without explanation, please revise and rewrite.

The manuscript was updated to include the abbreviations.

Introduction:

a. The introduction follow a purposeful structure and content.

The introduction was revised.

b. Please review the results of several other studies in the introduction and explain the difference between this study and the others.

The introduction part was improved with revised paragraphs and additional references were added.

c. Please move the explanation of EDTA from paragraph 6 to paragraph 4 (line 1).

The explanation was moved following the reviewer’s suggestion.

Materials and methods:

a. The thickness of the layers (coatings) is not specified. Please, give more information about the case.

Information is contained in the following sentence: „Tensile tests were carried on rectangular specimens (50mm × 8mm × approx. 0.1mm), at 5mm/min.

b. Why were these contents (0.5 and 1) chosen from Glt and Alg?

The concentrations chosen for Glt or Alg were incrementally increased : 0%, 0,5% and 1%, to evaluate their influence on the performances of the coatings. We have stopped at 1% because the coatings became brittle.

c. Please move Figures 2 and 4 to the Supporting Information.

Since Figures 1 – 4 can be better understood together, we have moved them all to Supporting info.

Thermal section:

a. The results obtained in this section are very interesting, but unfortunately, there has been no discussion and it is only a data report, but these results can be used in the best way. I suggest that describe the mechanism of performance of the materials carefully.

The TGA results discussion was improved following the reviewer’s suggestion.

  1. In Figure 7b, the curves of samples cannot be distinguished from each other at all. Show the curves in distinct colors that are recognizable.

Figure 7 (Figure 3 now) was modified (different colors were already used to distinguish the samples and we replaced dot-lines with solid lines), in accordance with the reviewer’s suggestion.

  1. It is better to show the curves up to a temperature of 1000 °C so that the stabilization of the curves can be seen (especially the DS5 and DS6 curve).

The DS5 and DS6 contain alginate which leads to the formation of Na2CO3 which undergoes thermal degradation at high temperatures (we have performed the TGA up to 900°C, since the maximum for our equipment is 1000°C). There is not a significant information to be gained if the range is increased by 100°C. All weight loss domains are already present in the analysis for the selected range (25-900°C).

d. List all temperature ranges in a table and move Figure 7 to the Supporting Information.

A table containing the information regarding the temperatures was added to the manuscript, but we have opted to keep Figure 7 (Figure 3 now) in the manuscript for a clearer presentation of the results.

  1. DSC analysis section:  
    Please remove the sentence "Additionally, DSC analysis was performed to obtain complementary data regarding the thermal properties of the nanocomposite coatings. Thus, DSC thermograms are comparatively illustrated in Figure 8." and replace it with this sentence: "DSC method was used for evaluating the thermophysical properties of samples as shown in Figure 8."

We have modified the sentences following the reviewer’s observation.

f. In figure 8, vertical axis, please indicate the direction of exothermicity.

We have updated Figure 8 (Figure 3 now)  and introduced the exo up sign.

g. In Figure 8, DS6 curve, what is the reason for the peak appearing at 50°C?

For DS6 the signal at around 50°C is an endothermic relaxation event, that can be attributed to the Tg of gelatin polymer.

  1. In figure 8, DS5 curve, what is the reason for the peak appearing in the range of 0 to 25°C?

The analyses were repeated, and the second cycle was used to remove the thermal stress in the material which was responsible for the initial signals present on the original graphs.

i. In figure 8, DS2 curve, what is the reason for the peak appearing in the range of -10 to -25°C?

The analyses were repeated and the second cycle was used to remove the thermal stress in the material which was responsible for the initial signals present on the original graphs.

  1. Why the peaks that were asked in the previous three comments have appeared in the same curves and are not seen in the rest of the curves?

This is probably due to alginate PVA interaction. The other samples did not display the same residual thermal stress.

k. List all temperature ranges in a table and move Figure 8 to the Supporting Information.

DSC plots were moved to Supporting info, and we added Table 4 (including the Tg values for the samples) to the manuscript.

Mechanical properties:

a. One of the most important sections of this manuscript is the mechanical properties. Important factors can be reported from the stress-strain curve, such as elongation at break, strength, and elastic modulus. Please report all these factors in a table and move Figure 9 to the Supporting Information. By presenting a table, comparison of samples will be done much better, and more accurate information will be given to the reader.

Table 5 was added, but we decided to keep the figure in the manuscript for a more facile comparison of the results.

Decontamination section:

a. Please improve the discussion.

The discussion was expanded.

b. The quality of the Figure 11 is very poor and it is better to improve the quality.

Figure 11 (Figure 6 now) was modified.

Reviewer 2 Report

The subject of peelable coatings on decontamination is a very good choice and important in the functions of radiation laboratory. The choice of wording “eco-friendly” is somewhat poor since considerations of dealing with the contamination were limited to ‘in a laboratory’ only and some further thinking on what happens to the waste after when it is removed from the lab could have been done also.

The biggest shortcoming of the manuscript is the use of the word decontamination factor (DF) too freely. DF is typically the ratio of activity before and after the treatment but here also multiplication of 100 was used which confuses the reader as the numbers are only glanced and the text is not carefully read.

Below I give some questions and suggestions that should be addressed before the manuscript can be considered for publication.

Page 2:

1) I strongly suggest using IUPAC units Bq instead of Ci (100microcuries)

2) “..a removal efficiency of only 56.96% for 137Cs” –

Here a removal efficiency is given that could be used also later instead of the decontamination factor that actually is the DF percentage

3) “…containing Fe3O4 as magnetic absorbent for 137Cs, obtaining a maximal removal efficiency from paint-coated cement of approximately 96%.”

The reader can get the wrong impression that Fe3O4 is a good Cs-adsorbent even though it is not.

 Page 6

DF decontamination factor is the activity ratio before and after and here is a percentage that is 100 higher than typical DF and so makes confusion please change this for example what you have used Decontamination efficiency DE

Results and discussion a)

                           Should the authors continue their tough pattern beyond 'in-house' problems and consider also how the 'eco-friendly' materials could be disposed off permanently? Incineration, final disposal in where?? and what are the benefits of using less PVA then? In the production of the chemical or what??

Page 8 el).”… For DS6 the peak maxima situated around 3291cm-1 could be attributed to N-H stretching in gelatine.”

                           This N-H stretch can not be seen in the IR, actually a large part of IR spectra could be just water (3260, 1635)??

Page 9.

Is there any logical change in the TG based on the DS composition - not seen in figure 7a?

Figure 7 b is not visible

Page 10

Interestingly the DSC in figure 8 starts from -25C?, and no typical DSC for bentonite is observed? perhaps too low quantity?

The tensile tests is the best bit of the study and also the 2. important after the uptake-properties - how this relates to the 'ecological thinking' of the materials

Page 11

. “The necessity to minimize the exposure to radionuclides of the personnel conducting the experiments was another factor that influenced the choice to test only four decontamination solutions.”

                           Perhaps this is not a good sentence here because it can question the whole study

).” Yttrium-90 is a therapeutic radionuclide of considerable interest, and it is being used in a variety of well-known radiopharmaceuticals44. 90Sr-90Y generator is the standard source45 when preparing radiopharmaceuticals for radionuclide therapy, ensur-ing long-term, continuous availability of no-carrier-added 90Y. However, prior to patient administration, it is always necessary estimating the purity of 90Y solution (90Sr impuri-ties). Americium-241 is an α-emitter with a low γ-ray byproduct. γ emission of 241Am are useful in passive diagnosis of thyroid function46. Even if for diagnosis applications 241Am is used in encapsulated pellets46, in general-purpose radioisotope laboratories contamination with actinides is still a matter of concern 47.”  

                           Not relevant here? Omit??

Page 12

“The targeted surface may have been slightly "in-depth" contaminated due to the acid, but the decontaminating formulas still performed well under these circumstances.”

                           Actually this could be the other way around since the low pH may prevent the formation of bonds between the material and radionuclide and if in-depth refers to possible degradation of the surface that is very small in comparison to the layer of the desorbent.

Author Response

Additional explanations were added in the introduction section following the reviewer’s suggestion.

The biggest shortcoming of the manuscript is the use of the word decontamination factor (DF) too freely. DF is typically the ratio of activity before and after the treatment but here also multiplication of 100 was used which confuses the reader as the numbers are only glanced and the text is not carefully read.

Information was corrected, and supplementary explanations were added.

Below I give some questions and suggestions that should be addressed before the manuscript can be considered for publication.

Page 2:

1) I strongly suggest using IUPAC units Bq instead of Ci (100microcuries)

The manuscript presentation was improved to display Bq units.

2) “..a removal efficiency of only 56.96% for 137Cs” –

Here a removal efficiency is given that could be used also later instead of the decontamination factor that actually is the DF percentage

This sentence was revised following the reviewer’s suggestion.

3) “…containing Fe3O4 as magnetic absorbent for 137Cs, obtaining a maximal removal efficiency from paint-coated cement of approximately 96%.”

The reader can get the wrong impression that Fe3O4 is a good Cs-adsorbent even though it is not.

This sentence was revised.

 Page 6

DF decontamination factor is the activity ratio before and after and here is a percentage that is 100 higher than typical DF and so makes confusion please change this for example what you have used Decontamination efficiency DE

We admit that we might have created confusion, the presentation was improved to include the decontamination efficiency instead of DF.

Results and discussion a)

 Should the authors continue their tough pattern beyond 'in-house' problems and consider also how the 'eco-friendly' materials could be disposed off permanently? Incineration, final disposal in where?? and what are the benefits of using less PVA then? In the production of the chemical or what??

The manuscript presentation was improved to include additional explanations.

Page 8 el).”… For DS6 the peak maxima situated around 3291cm-1 could be attributed to N-H stretching in gelatine.” This N-H stretch can not be seen in the IR, actually a large part of IR spectra could be just water (3260, 1635)??

This sentence was revised.

Page 9.

Is there any logical change in the TG based on the DS composition - not seen in figure 7a?

Additional explanations were added in the manuscript.

Figure 7 b is not visible

Figure 7 (Figure 3 now) was modified

Page 10

Interestingly the DSC in figure 8 starts from -25C?, and no typical DSC for bentonite is observed? perhaps too low quantity?

Additional explanations were added in the manuscript.

The tensile tests is the best bit of the study and also the 2. important after the uptake-properties - how this relates to the 'ecological thinking' of the materials

Explanations were added.

Page 11

. “The necessity to minimize the exposure to radionuclides of the personnel conducting the experiments was another factor that influenced the choice to test only four decontamination solutions.” Perhaps this is not a good sentence here because it can question the whole study

 This sentence was removed.

).” Yttrium-90 is a therapeutic radionuclide of considerable interest, and it is being used in a variety of well-known radiopharmaceuticals44. 90Sr-90Y generator is the standard source45 when preparing radiopharmaceuticals for radionuclide therapy, ensur-ing long-term, continuous availability of no-carrier-added 90Y. However, prior to patient administration, it is always necessary estimating the purity of 90Y solution (90Sr impuri-ties). Americium-241 is an α-emitter with a low γ-ray byproduct. γ emission of 241Am are useful in passive diagnosis of thyroid function46. Even if for diagnosis applications 241Am is used in encapsulated pellets46, in general-purpose radioisotope laboratories contamination with actinides is still a matter of concern 47.”  

                           Not relevant here? Omit??

 This paragraph was removed.

Page 12

“The targeted surface may have been slightly "in-depth" contaminated due to the acid, but the decontaminating formulas still performed well under these circumstances.”

                           Actually this could be the other way around since the low pH may prevent the formation of bonds between the material and radionuclide and if in-depth refers to possible degradation of the surface that is very small in comparison to the layer of the desorbent.

This paragraph was removed.

Additional explanations were added in the introduction section following the reviewer’s suggestion.

The biggest shortcoming of the manuscript is the use of the word decontamination factor (DF) too freely. DF is typically the ratio of activity before and after the treatment but here also multiplication of 100 was used which confuses the reader as the numbers are only glanced and the text is not carefully read.

Information was corrected, and supplementary explanations were added.

Below I give some questions and suggestions that should be addressed before the manuscript can be considered for publication.

Page 2:

1) I strongly suggest using IUPAC units Bq instead of Ci (100microcuries)

The manuscript presentation was improved to display Bq units.

2) “..a removal efficiency of only 56.96% for 137Cs” –

Here a removal efficiency is given that could be used also later instead of the decontamination factor that actually is the DF percentage

This sentence was revised following the reviewer’s suggestion.

3) “…containing Fe3O4 as magnetic absorbent for 137Cs, obtaining a maximal removal efficiency from paint-coated cement of approximately 96%.”

The reader can get the wrong impression that Fe3O4 is a good Cs-adsorbent even though it is not.

This sentence was revised.

 Page 6

DF decontamination factor is the activity ratio before and after and here is a percentage that is 100 higher than typical DF and so makes confusion please change this for example what you have used Decontamination efficiency DE

We admit that we might have created confusion, the presentation was improved to include the decontamination efficiency instead of DF.

Results and discussion a)

 Should the authors continue their tough pattern beyond 'in-house' problems and consider also how the 'eco-friendly' materials could be disposed off permanently? Incineration, final disposal in where?? and what are the benefits of using less PVA then? In the production of the chemical or what??

The manuscript presentation was improved to include additional explanations.

Page 8 el).”… For DS6 the peak maxima situated around 3291cm-1 could be attributed to N-H stretching in gelatine.” This N-H stretch can not be seen in the IR, actually a large part of IR spectra could be just water (3260, 1635)??

This sentence was revised.

Page 9.

Is there any logical change in the TG based on the DS composition - not seen in figure 7a?

Additional explanations were added in the manuscript.

Figure 7 b is not visible

Figure 7 (Figure 3 now) was modified

Page 10

Interestingly the DSC in figure 8 starts from -25C?, and no typical DSC for bentonite is observed? perhaps too low quantity?

Additional explanations were added in the manuscript.

The tensile tests is the best bit of the study and also the 2. important after the uptake-properties - how this relates to the 'ecological thinking' of the materials

Explanations were added.

Page 11

. “The necessity to minimize the exposure to radionuclides of the personnel conducting the experiments was another factor that influenced the choice to test only four decontamination solutions.” Perhaps this is not a good sentence here because it can question the whole study

 This sentence was removed.

).” Yttrium-90 is a therapeutic radionuclide of considerable interest, and it is being used in a variety of well-known radiopharmaceuticals44. 90Sr-90Y generator is the standard source45 when preparing radiopharmaceuticals for radionuclide therapy, ensur-ing long-term, continuous availability of no-carrier-added 90Y. However, prior to patient administration, it is always necessary estimating the purity of 90Y solution (90Sr impuri-ties). Americium-241 is an α-emitter with a low γ-ray byproduct. γ emission of 241Am are useful in passive diagnosis of thyroid function46. Even if for diagnosis applications 241Am is used in encapsulated pellets46, in general-purpose radioisotope laboratories contamination with actinides is still a matter of concern 47.”  

                           Not relevant here? Omit??

 This paragraph was removed.

Page 12

“The targeted surface may have been slightly "in-depth" contaminated due to the acid, but the decontaminating formulas still performed well under these circumstances.”

                           Actually this could be the other way around since the low pH may prevent the formation of bonds between the material and radionuclide and if in-depth refers to possible degradation of the surface that is very small in comparison to the layer of the desorbent.

This paragraph was removed.

Round 2

Reviewer 1 Report

Thank you for the changes in the manuscript. Accept as it.